# Tools for Assessing Knowledge of Back Health in Adolescents: A Systematic Review Protocol

**DOI:** 10.3390/healthcare10081591

**Published:** 2022-08-22

**Authors:** Adriana Cristina Fiaschi Ramos, Regina Márcia Ferreira Silva, Thailyne Bizinotto, Leonardo Mateus Teixeira de Rezende, Vicente Miñana-Signes, Manuel Monfort-Pañego, Priscilla Rayanne e Silva Noll, Matias Noll

**Affiliations:** 1Campus Goiânia, Federal University of Goiás, Goiânia 74690-631, Brazil; 2Campus Morrinhos, Federal Institute Goiano, Morrinhos 74605-050, Brazil; 3Campus Ceres, Federal Institute Goiano, Morrinhos 74605-050, Brazil; 4Campus Valencia, University of Valencia, 46010 Valencia, Spain; 5Campus São Paulo, University of São Paulo, São Paulo 01246-904, Brazil

**Keywords:** back pain, spine, postural habits, questionnaire, validation

## Abstract

Back pain is common in adolescents as a result of their typical daily activities. There is a critical need for developing instruments that can assess the adolescents’ knowledge of proper posture, because adequate postural habits are essential for preventing back pain and facilitating physical well-being. Unfortunately, there is insufficient understanding about appropriate back health in the general public, even though this knowledge is decisive in the development of physical skills and attainment of health literacy. Furthermore, relevant substantive literature is scarce. Therefore, the proposed systematic review aims to identify instruments that are used for assessing knowledge of back health in adolescents. Relevant search terms and descriptors will be combined, and searches will be carried out in a uniform sequence within the PubMed, Embase, CINAHL, and Cochrane Library databases. Eligible articles must present data on the assessment of the adolescents’ knowledge of back health and describe the applied instrumentation. Articles will be selected by two reviewers independently; all disagreements will be resolved by a third reviewer. Mendeley and the Rayyan software will be used for the systematic review, and the checklist proposed by Brink and Louw will be used to verify the methodological quality of the included studies. Our findings may confirm the relevance of constructing and validating back health instruments for use in Brazil and other countries.

## 1. Introduction

Since pain is a very broad concept, it is challenging to define effectively and succinctly. Nevertheless, pain can be defined as an experience of unpleasant sensations and emotions that are associated with actual or potential injuries [1]. The definition of pain also encompasses an experience that makes it impossible for an individual to perform daily activities for at least 1 day [2]. Pain is a limitation in all fields of life, including its social, interpersonal, and economic domains [3]. Studies have reported a considerable increase in the incidence of back pain in adolescents (age: 10–19 years) [4,5]. A Danish survey on 1389 adolescents (between 13 and 16 years of age) revealed that the incidence of pain in this population increased by 19.4% in recent years [6]. Possible causes of this increase include daily activities, such as television viewing and use of cell phones, notebooks, and tablets [3,7,8].

Back pain is relevant to postural habits and activities of daily living (ADLs) [9]. Daily activities that are associated with suboptimal postures can trigger nonspecific musculoskeletal disorders and compromise the blood flow [10]; both of these can then lead to the onset of pain [9].

This is a critical issue, because pain that begins in adolescence can persist well into adulthood [11,12]. In adolescents, complaints of back pain are recurrent and the pain itself affects more than one region of the spine [4]. A probable cause for this is idiopathic scoliosis, which is characterized by the lateral curvature of the spine; this condition can persist even after the affected individual has attained adulthood [13]. Treatment can be initiated through the use of an orthosis, which is a rigid thoracolumbar brace [13]. 

Several factors influence the onset of back pain in adolescents; these include the quality and quantity of sleep, performance of regular physical exercise, age, sex, parental education level, genetic predisposition, and inappropriate postural behavior [4,11]. Therefore, it is important that this vulnerable demographic be administered education that allows a comprehensive understanding of the need to adopt adequate postural habits in the execution of daily tasks [14].

The typical activities of adolescents include the use of smartphones, tablets, and notebooks. Furthermore, they are excessively exposed to televisions and computer screens [4,14]. Adolescents also often carry heavy backpacks in an improper way, and this may unfavorably impact normal spinal development [15,16,17]. Adolescents also spend a considerable portion of their day attending classes at school, where they may sit in ergonomically inappropriate furniture; this also contributes to an overload on the back (overall and, particularly, with the adoption of an improper posture) [9,14]. Moreover, obese and sedentary adolescents are more likely to experience back pain than their non-obese or non-sedentary peers [7,18].

Evidence-backed and accurate knowledge regarding back health is essential for avoiding an inappropriate posture and consequent pain. Previous research has shown that even elementary school students develop a greater sense of responsibility for their own health once they understand how body movement is processed [19]. Thus, in order for individuals to change their lifestyle habits, they first need to understand how these habits impact their health. From an awareness of posture and its impact on the quality of life, individuals of all ages can begin to correct postural habits that harm their body [20]. Therefore, relevant information must be accessible, because this enables the beginning of appropriate health education, effective learning, and establishment of healthy habits [19]. Researchers believe that when students are provided with rigorous and evidence-based knowledge about back health and the factors influencing back pain, the probability of injury prevention in this demographic increases [21,22].

In this context, educational institutions can play a fundamental role in informing adolescents about appropriate back health [23]. For example, the inclusion of educational programs in the general curriculum that are aimed at teaching students how to perform simple movements while respecting proper postural positioning has reportedly reduced the incidence of motor limitations [21,24]. Most of the posture-related research on existing school curricula is based on currently implemented practices and how these impact the students’ health. This knowledge is essential, and several of the interventions that have been administered in the context of health education and learning have been evaluated through formal tests or questionnaires; these evaluations are aimed at assessing the teachers’ and students’ degree of knowledge on this topic [22].

The organization of health education programs differs across institutions and settings; varying emphasis has been placed on the knowledge of correct postural habits, practice of physical exercises, and training through practical experiences, lectures, films, and meetings [25,26]. In order for a greater number of students to benefit from this type of programming, researchers recommend that relevant programs be implemented in the school curriculum from as early as when the children are in elementary schools; this would allow an adequate temporal verification of the curriculum’s efficiency and effectiveness in improving the back health of adolescents [27,28]. A Hungarian study recommended that such programming be implemented in children as young as 4 years of age [29]. However, for this to be possible, it is necessary to establish guidelines specific to these educational programs, because the immense variety of currently available and applied programs makes it difficult to implement optimal curricula in the school environment [30].

There is also a need to rigorously measure the adolescents’ and teachers’ knowledge of the impact of self-care on back health through the use of self-reported instruments on postural habits and back pain (i.e., questionnaires). However, it is essential that these instruments are standardized thoroughly [31]. The lack of standardization of the currently used instrumentation, in addition to the scarcity of research-backed tools, has resulted in relatively low-quality evidence on this topic [27,32,33].

Good quality data are obtained through the use of validated and reliable instruments, which help avoid erroneous findings that compromise the entire research [33]. Optimal instruments can be obtained through the accurate translation, cross-cultural adaptation, and validation of existing instruments or by creating new, context-specific instruments. These processes require the adoption of rigorous methodology, especially considering the many alternative approaches for the itemization and review of published and grey literature that are available [34,35]. It is important to consider several aspects, such as the instruments that can be used to assess the level of knowledge that people have on back health and the descriptive characteristics of these instruments. Few self-reported tools that assess knowledge regarding back health in adolescents are available; the Postural Habits Related to Back Health in Daily Activities Questionnaire and the Back-care Behavior Assessment Questionnaire are two such tools [10,13].

Therefore, the aims of the proposed systemic review are to analyze the currently available instruments that assess the knowledge of back health among adolescents and the methodological quality of the studies that have used these instruments to date.

## 2. Materials and Methods

The present protocol was prepared in accordance with the *Preferred Reports for Systematic Reviews and Meta-analyses* (PRISMA-P) guidelines [36]. This systematic review was registered in the International Registry of Prospective Systematic Review (registration number: CRD 42022347726) [37]. With the publication of this protocol, the authors aim to increase transparency and reproducibility and prevent a duplication of efforts.

### 2.1. Research Question

The following study questions will be examined in the proposed systematic review:(1)What instruments are currently available for assessing the knowledge of back health in adolescent students?(2)What is the methodological quality of the studies that have used these instruments?

### 2.2. Eligibility

#### Inclusion and Exclusion Criteria

This systematic review will consider relevant articles with no restrictions on the language or publication date. For eligibility, articles must meet all of the following inclusion criteria: (a) they detail studies that have included instruments that are used to assess knowledge of back health, postural habits, and awareness of body movement; (b) they detail studies that have used instruments that have been (preferably) validated or, at least, tested for their reproducibility; and (c) they detail studies that have enrolled adolescent students aged 14–17 years.

The following articles will be excluded from the proposed systematic review: (a) previous systematic reviews, opinion articles, case studies, and reports; (b) articles describing other aspects of health knowledge (i.e., studies not evaluating back health); and (c) articles including participants outside the target age range or those with physical and/or mental impairments (the latter criterion has been established to allow for greater generalizability). A list of the excluded articles, with the reasons for their exclusion, will be included in the systematic review as a Appendix A.

### 2.3. Database Search

Four categories of search strategies have been created for the proposed systematic review. Each category contains specific search terms, namely “instrument”, “knowledge”, “back”, and “adolescents”. 

The search terms and descriptors will be combined, and two reviewers will independently perform searches in a uniform sequence within the PubMed, Embase, CINAHL, and Cochrane Library databases. A third reviewer will check for inconsistencies between the findings of the two original reviewers and will resolve any identified disagreements. 

The bibliography of the identified articles will also be searched as a complementary search, and the manual searches will also be conducted using Google Scholar (https://scholar.google.com.br) as an additional search. Table 1 shows the logical structure of the search strategy and all the keywords and Boolean Operators that will be used to search the four databases. The search strategy itself is presented as material supplementary to this article (Appendix A).

### 2.4. Review Process

The articles identified after searching the databases will be imported into Mendeley (Mendeley Desktop 1.19.8, London, UK) a bibliographic reference managing software (Mendeley, London, UK), which will identify and remove duplicates. Using the Rayyan software [38], the reviewers will then perform article selection by reading the titles and abstracts of the identified articles; they will evaluate each article for conformance with the inclusion criteria. Thereafter, the full text of the selected articles will be read to confirm their eligibility for inclusion in the systematic review, as described in Section 2.5 [38].

### 2.5. Article Selection

As mentioned previously, the selection of eligible articles for inclusion in the systematic review will be performed independently by two reviewers; a third senior reviewer will resolve any disagreements. The Cohen’s Kappa coefficient and percentage agreement will be used to verify inter-evaluator reliability for the classification of the individual components [39]. The flowchart for article selection for this systematic review is provided in Figure 1.

### 2.6. Reviewer Training

Reviewers participating in this systematic review must be aware of the inclusion and exclusion criteria. To this end, the reviewers will complete training that involves the reading of 50 titles and abstracts of preselected articles in order to verify their eligibility [40]. Then, the reviewers will train in the use of tools for methodological analysis, including Mendeley and Rayyan [38].

### 2.7. Data Extraction

The following data will be extracted from the eligible articles: study, title, authors, place of study, year of publication, type of study, instrument used for assessment, validation of said instrument, information on reliability and reproducibility, information on the employed method, and number of boys and girls. Table 2 is a template that will be used for presenting the data extracted from the articles identified in the proposed systematic review. This systematic review will be performed by analyzing data collected from previously published articles (i.e., secondary data); therefore, it will be exempt from ethical approval. Once the systematic review is completed, it will be submitted for publication.

### 2.8. Methodological Quality

The methodological quality of the identified articles will be evaluated using a checklist developed by Brink and Louw [41]. This tool, which is composed of 13 items, assesses the reliability and validity of the selected instrument (either separately or together) [41]. These 13 items are listed below: 

Item 1:If human subjects were used, did the authors give a detailed description of the sample of subjects used to perform the (index) test on?Item 2:Did the authors clarify the qualification or competence of the examiner(s) who performed the test? Item 3:Was the reference standard explained? Item 4:If interrater reliability was tested, were raters blinded to the findings of other raters?Item 5:If intra-rater reliability was tested, were the raters blinded to their own previous findings for the test under assessment? Item 6:Was the test order varied? Item 7:Was the time period between testing and retesting short enough for the researchers to be confident that the response would not have changed between the two tests? Item 8:Was the stability (or theoretical stability) of the variable being measured taken into account when determining the suitability of the time interval between repeated measures?Item 9:Were the standard reference evaluations and testing performed independently? Item 10:Was the test run described in sufficient detail to allow for reproducibility? Item 11:Was the reference evaluation described in sufficient detail to allow for reproducibility of the investigation? Item 12:Was the sample composition presented within the study description? Item 13:Were the statistical methods appropriate for the purpose of the study? 

These 13 items can be categorized more generally as follows: items related to the assessment of reliability and validity (*n* = 5), items related only to the assessment of reliability (*n* = 4), and items related only to the assessment of validity (*n* = 4). The responses to each item are “yes”, “no”, and “not applicable” [34].

The methodological quality of the articles included in the study will be determined by the two aforementioned reviewers. If there are disagreements, they will be resolved by a third reviewer. The Cohen’s Kappa coefficient [39] will be used to measure the reliability between evaluators as well as the inter-evaluator agreement.

### 2.9. Summary of Evidence 

A systematic review consists of a survey of previously published studies, and similar articles are grouped together. Then, a critical analysis of the methodology used in these studies is performed. It is from this investigation that the evidence necessary for the development of a systematic review is consolidated [40].

Data (e.g., authors, year of publication, title, location of the study, type of study, instrumentation used for assessment, validation of said instrumentation, and the method developed in the study) will be evaluated according to the abovementioned protocols. Following this, there will be an assessment of the previously published studies in terms of the instruments used to measure the knowledge of back pain among adolescents. The checklist developed by Brink and Louw will allow for the assessment of the methodological quality of the included studies [41].

## 3. Expected Results

This study will search for publications on instruments evaluating the knowledge of back health among adolescents and will also formally evaluate the methodological quality of the selected studies. Being aware of the instruments validated for this purpose is of paramount importance, because such instruments can help us gain an understanding of the postural problems in adolescents, prevent a deterioration in their back health, and encourage the development of effective and rigorous future studies and clinical/educational practices on this topic.

## 4. Discussion

The proposed systematic review aims to rigorously and comprehensively describe the existing literature on self-reported instruments that are currently used to estimate postural knowledge among adolescents. The findings of our review can assist in the development of a better and more efficient evidence base on postural educational programs that can be included in the existing school curricula in Brazil and other countries (e.g., in physical or health education classes). Such educational programming aims to optimize the adolescent students’ awareness of their own body, thereby decreasing the incidence of back pain and consequent absenteeism [21,42,43] and preventing both short-term and long-term adverse health effects in this population. Studies have reported that implementation of such educational programs improved the awareness of body movements among adolescents, which in turn led to a decrease in the risk behaviors for poor back health [44,45]. Moreover, this systematic review is expected to (1) provide a comprehensive evidence base for improving the existing instruments or for creating new, more efficient tools that expand the framework of possibilities in this area and (2) demonstrate optimization of programs with a demonstrable public health and preventive medicine impact. 

### Strengths and Limitations

Notable strengths of the present study include the lack of limitations on the language and year of publication in the search strategy, inclusion of an analysis of the methodological quality of the included studies, strict adherence to ethical practices, and appropriate disclosures of competing interests (if any). In summary, the proposed review can confirm the relevance of the construction and validation of instruments for use in the planning of postural programs for inclusion in the educational curricula in Brazil and other countries, with the aim of achieving population health advancement and short- and long-term prevention of the adverse effects of poor posture on health. Therefore, future studies are required to promote the importance of this area of research.

The proposed study has certain limitations as well. First, the concepts of back-health knowledge, postural habits, and other relevant health terminology are not standardized. Moreover, it is possible that this systematic review will verify a lack of suitable instruments for evaluating the adolescents’ knowledge of correct postural habits in terms of their typical daily activities; if no study meets the specified eligibility criteria, this research will be considered an empty review [46]. For future studies, there needs to be homogeneity in the conceptual terminology of the knowledge of spine health and postural habits. Furthermore, development of instruments that can be used to analyze the knowledge of spinal health in adolescents is required. 

Despite the relative scarcity of data, this literary survey is extremely important in terms of improving the knowledge of instruments that are used for assessing postural habits among adolescents [27,31,32]. This knowledge can be used to promote the implementation of educational programs aiming to minimize the incidence of back pain in this population, which would ultimately improve their quality of life.

## Figures and Tables

**Figure 1 healthcare-10-01591-f001:**
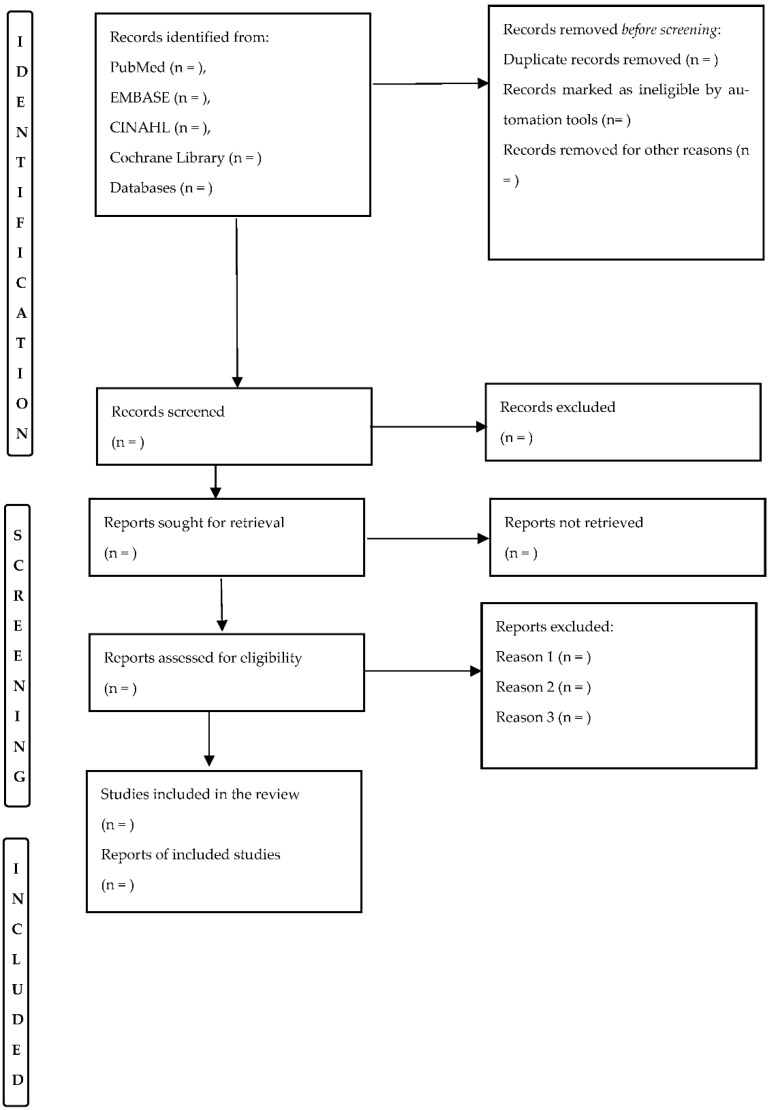
Selection of articles for inclusion in the systematic review (PRISMA-P).

**Table 1 healthcare-10-01591-t001:** Search strategy for the databases.

Search Strategy	
1	“surveys and questionnaires” OR “health care surveys” OR “instrument” OR “form” OR “survey” OR “questionnaire” OR “measurement” OR “tool” OR “assessment” OR “score” OR “self-report” AND
2	“knowledge” OR “health knowledge” OR “attitude” OR “practice” OR “posture knowledge” OR “knowledgeability” OR “knowledgeably” AND
3	“back” OR “back pain” OR “posture” OR “spine” OR “back posture” OR “back health” OR “posture habits” OR “spine health” OR “spine posture” OR “spine care” AND
4	“adolescent” OR “students” OR “teen” OR “teenager” OR “youth” OR “juvenile” OR “school enrollment” OR “enrollment school” OR “young” OR “minor”

**Table 2 healthcare-10-01591-t002:** Summary of data extracted from the identified articles.

Study	Title	Author	Year of Publication	City	Sex	Study Type	Assessment Instrument	Instrument Validation	Method	Reliability and Reproducibility



## Data Availability

Not applicable.

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
