# Peer review of "Tools for Assessing Knowledge of Back Health in Adolescents: A Systematic Review Protocol"

_healthcare, 2022, doi:10.3390/healthcare10081591_

Round 1

Reviewer 1 Report

My humble consideration as a reviewer is that a systematic review protocol does not add any new scientific knowledge. It simply describes how it is going to be done, and for this it would suffice to register PROSPERO, where said protocol will appear. Also, this introduction could be used in the review and not here in this protocol.

Methods present flags.

-Why authors do not mencionate PRISMA or Cochrane Handbook?

-SCOPUS and Web of Science would be included as first source, not as additional.

-Line 20: Please write correctly CINAHL Complete.

-Please use tags for PubMed search strategy.

-Why authors do not include meta-analysis?

Thanks.

Reviewer 2 Report

Dear authors apart from the methods followed, also try to include the list of reasons for which the articles would be excluded from the full-text review 

Also, mention the data extraction fields in the included table for data collection 

Reviewer 3 Report

The manuscript is a systematic review study aimed to identify instruments assessing knowledge regarding back health in adolescents. earches will be carried out in the PubMed, EMBASE, CINHAL and Cochrane Library databases. Eligible articles must present data regarding assessment of knowledge on back health and describe instrumentation. Article selection will be conducted independently by two reviewers, with disagreements resolved by a third reviewer. Mendeley and Rayyan software will be employed for the systematic review, and the instrument proposed by Brink and Louw will be used to verify the methodological quality of the articles

I read the article with interest, the title is well thought out and faithfully reflects the content of the study.

The abstract is adequately developed. In the introduction, the characteristics of the low back pain have been described. In materials and methods are adequately developed. The discussion is sufficiently developed.

Nevertheless, some major changes are needed to be considered suitable for publication.

Comment 1: In the Abstract: It would be appropriate to subdivide into sub-paragraphs: backgounds, methods, results, conclusion.

Comment 2: In the introduction: It would be better to elaborate on the aspects on the pathogenesis of the adolescent idiopathic scoliosis as a cause of low back pain, adding appropriate bibliographical references. For example: (Di Maria F, et al. (2021) "Immediate Effects of Sforzesco® Bracing on Respiratory Function in Adolescents with Idiopathic Scoliosis. Healthcare (Basel)").

Comment 3: In the introduction: It would be appropriate to add brief notes on the treatment, adding appropriate bibliographical references.

Comment 4: In materials e method: Please specify how many of the participants were female, how many were male, and if there was any heterogeneity in the data.

Comment 5: In materials e method:  The PRISMA according to systematic review study should be added. It is also appropriate to add a summary of authors, year of publication and a summary of the study to be clearer to the reader.

Comment 6: In the discussion: It's not very clear what should be done in future studies to improve the limitations of your study.

Comment 7: Finally, additional English editing is needed. The Non-Native Speakers of English Editing Certificate was not signed.

Round 2

Reviewer 1 Report

Thank you. However, I consider that publish a protocol review does not provide knowledge respect previous reviews. I encourage to add the information content in this manuscript in the review when it was made and published.

Reviewer 3 Report

Thank you for carrying out the required reviews, the article is now suitable for publication.